# 3D virtual pathohistology of lung tissue from Covid-19 patients based on phase contrast X-ray tomography

**Marina Eckermann[1,2†], Jasper Frohn[1†], Marius Reichardt[1†], Markus Osterhoff[1], Michael Sprung[3], Fabian Westermeier[3], Alexandar Tzankov[4], Christopher Werlein[5,6], Mark Kühnel[5,6], Danny Jonigk[5,6*], Tim Salditt[1,2*]**

[1]Institut für Röntgenphysik, Georg-August-Universität, Göttingen, Germany; [2]Cluster of Excellence 'Multiscale Bioimaging: from Molecular Machines to Networks of Excitable Cells' (MBExC), University of Göttingen, Göttingen, Germany; [3]Deutsches Elektronen-Synchrotron (DESY), Hamburg, Germany; [4]Institut für Medizinische Genetik und Pathologie, Universitätsspital Basel, Basel, Switzerland; [5]Medizinische Hochschule Hannover (MHH), Hannover, Germany; [6]Deutsches Zentrum für Lungenforschung (DZL), Hannover (BREATH), Germany

**\*For correspondence:**
Jonigk.Danny@mh-hannover.de (DJ);
tsaldit@gwdg.de (TS)

[†]These authors contributed equally to this work

**Competing interests:** The authors declare that no competing interests exist.

**Abstract** We present a three-dimensional (3D) approach for virtual histology and histopathology based on multi-scale phase contrast x-ray tomography, and use this to investigate the parenchymal architecture of unstained lung tissue from patients who succumbed to Covid-19. Based on this first proof-of-concept study, we propose multi-scale phase contrast x-ray tomography as a tool to unravel the pathophysiology of Covid-19, extending conventional histology by a third dimension and allowing for full quantification of tissue remodeling. By combining parallel and cone beam geometry, autopsy samples with a maximum cross section of 8 mm are scanned and reconstructed at a resolution and image quality, which allows for the segmentation of individual cells. Using the zoom capability of the cone beam geometry, regions-of-interest are reconstructed with a minimum voxel size of 167 nm. We exemplify the capability of this approach by 3D visualization of diffuse alveolar damage (DAD) with its prominent hyaline membrane formation, by mapping the 3D distribution and density of lymphocytes infiltrating the tissue, and by providing histograms of characteristic distances from tissue interior to the closest air compartment.

## Introduction

Severe progression of the 2019 coronavirus disease (Covid-19) is frequently accompanied by the clinical acute respiratory distress syndrome (ARDS) and respiratory failure, an organ manifestation responsible for the majority of Covid-19 fatalities. Lung injury associated with ARDS can be readily detected by radiographic chest imaging and clinical computed tomography (CT), which have assisted the diagnosis and management of Covid-19 patients (*Lee et al., 2020*; *Shi et al., 2020*; *Chung et al., 2020*). Here, so-called peripheral lung ground-glass opacities are the main radiological hallmark of ARDS, and can be linked to the histological observation of diffuse alveolar damage (DAD) with edema, hemorrhage, and intraalveolar fibrin deposition (*Ackermann et al., 2020*). These findings have also been reported for infections by Middle East respiratory syndrome coronavirus (MERS-CoV), SARS-CoV, and influenza viruses. Distinctive features of pulmonary involvement of Covid-19 include severe endothelial injury associated with the presence of intracellular virions and inflammation, disrupted cellular membranes, as well as widespread thrombosis with microangiopathy. As reported in *Ackermann et al., 2020*, alveolar capillary microthrombi were found to be nine times as prevalent in patients with Covid-19 as in patients with the also very aggressive H1N1

influenza A virus, also referred to as swine flu. Importantly, in Covid-19 lungs, a specific variant of new vessel growth - intussusceptive angiogenesis - was significantly more prevalent, that is, 2.7 times as high as in lungs of patients with H1N1 influenza A.

Histomorphological assessment of formalin-fixed, paraffin-embedded (FFPE) tissue stained with haematoxylin and eosin still represents the gold standard in histological diagnostics of non-neoplastic lung diseases, including DAD and virus induced pneumonia. In order to unravel the corresponding pathophysiology of the lungs, digitalization, visualization and quantification of the morphological changes associated with Covid-19 represent a key challenge, and require both high resolution and the capability to screen larger volumes. For this reason, imaging the intricate three-dimensional (3D) tissue architecture of the lung and its pathological alterations on multiple length scales calls for 3D extensions of well-established histology techniques.

In this work, we want to demonstrate the potential of propagation-based phase contrast x-ray tomography as a tool for virtual 3D histology in general, and in particular for the histopathology of Covid-19. Our work is based on the assertion that integration of 3D morphological information with well-established histology techniques can provide a substantial asset for unraveling the pathophysiology of SARS-Cov-2 infections. To this end, we have collected x-ray tomography data from the same autopsies, which were previously studied by immunohistochemical analysis and measurements of gene expression (*Ackermann et al., 2020*). We ask in particular, whether DAD and the morphology of blood vessels can be visualized and quantified in 3D. This is a timely scientific question in Covid-19 research, especially in view of increased intussusceptive angiogenesis reported in *Ackermann et al., 2020*. Here we present first results obtained from the postmortem lung samples of six patients who succumbed from Covid-19. We exemplify the capability of this approach by 3D visualization of the DAD with hyaline membrane formation, by mapping the 3D distribution and density of angiocentric inflammation (perivascular T-cell infiltration), and by providing histograms of characteristic distances from the tissue interior to the closest air compartment.

In contrast to conventional histology based on thin sections, propagation-based x-ray phase-contrast tomography (PC-CT) offers a full 3D visualization with isotropic resolution and without destructive slicing of the specimen (*Ding et al., 2019*; *Dejea et al., 2019*; *Saccomano et al., 2018*; *Töpperwien et al., 2018*; *Khimchenko et al., 2016*). The interaction of x-rays with the object is described by the continuous complex-valued index of refraction $n(\mathbf{r}) = 1 - \delta(\mathbf{r}) + i\beta(\mathbf{r})$. Phase contrast capitalizes on the fact that for hard x-rays, the real-valued decrement $\delta$ is several orders of magnitude higher in soft biological tissues than $\beta$, which accounts for absorption (*Nugent, 2010*) and that small contrast levels can be reconstructed by propagation imaging even at low fluence (*Jahn et al., 2017*). Contrast is formed by transformation of the phase shifts into measurable intensity variations by self-interference of the exit wave during free-space propagation between sample and detector (*Paganin and Nugent, 1998*; *Cloetens et al., 1999*). However, problems with phase retrieval, phase wrapping, insufficient coherence, or on the contrary strong phase gradients can render its application challenging. For lung imaging it has already been demonstrated as an advantageous imaging technique up to macroscopic scales (*Parsons et al., 2008*; *Stahr et al., 2016*; *Morgan et al., 2020*), including live animal models of respiratory diseases. But in contrast to other full-field phase contrast techniques, e.g. based on grating interferometry or analyzer crystals, it can also reach a resolution below optical microscopy (*Khimchenko et al., 2018*). In this work we exploit both the high resolution capability of PC-CT and the fact that its phase sensitivity is high enough to probe the small electron density variations of unstained tissue, embedded in paraffin, ethanol, or even aqueous buffer (*Töpperwien et al., 2019*). However, this requires careful optimization of photon energy, illumination function, and phase retrieval algorithms, as detailed further below.

The 3D virtual pathohistology approach for Covid-19 presented here was realized by implementing a novel multi-scale phase contrast x-ray tomography concept, with dedicated x-ray optics and instrumentation. Overview and regions-of-interest (ROI) scans were recorded on the same paraffin-embedded sample, covering a maximum tissue cross section of 8 mm by stitching different tomograms, and with a minimum voxel size of 167 nm in certain ROIs. Scale-bridging and dynamic ROI selection in close spatial and temporal proximity was implemented with dedicated instrumentation the GINIX endstation of the beamline P10/PETRA III (DESY, Hamburg) (*Salditt et al., 2015*). Specifically, we combined two optical geometries, which has only been realized at different synchrotron beamlines before: (i) Parallel beam tomography, covering a large *field of view* (FOV), with a pixel size of 650 nm. In this setting, a volumetric throughput on the order of $10^7 \ \mu m^3/s$ was achieved, while

maintaining the ability to segment isolated cells in unstained tissue. (ii) Cone beam geometry for recording of highly magnified holograms, based on advanced x-ray waveguide optics, providing mode filtering, that is, enhanced spatial coherence and smooth wavefronts. Based on the geometrical magnification, the effective pixel size can be adjusted in the range 10 nm–300 nm. The two imaging schemes are shown schematically in *Figure 1c and d*, respectively. Further, using this particular optics, together with appropriate choices of photon energy and geometric parameters, we can reach extremely small Fresnel numbers $F$ of the deeply holographic regime, well below the typical range exploited at other nano-tomography instruments. This offers the advantage of highest phase sensitivity sufficient to even probe the small electron density variations of unstained tissue, at relatively low dose (*Hagemann and Salditt, 2018*). To exploit this sensitivity, we use advanced phase retrieval methods including non-linear generalizations of the CTF-method (*Cloetens et al., 1999*) based on Tikhonov regularization (*Lohse et al., 2020*).

## Results

In total, we investigated six postmortem lung samples from Covid-19 patients (*Menter et al., 2020*). A tissue micro-array paraffin block with samples of all six patients and the corresponding HE stain is shown in *Figure 1a*. Information about age, gender, hospitalization, clinical, radiological and histological characteristics of all patients are shown in Table 2. Before presenting the results for the six Covid-19 patients for each of the imaging levels (scales), we give an overview on the typical datasets for one exemplary sample (I), see *Figure 2*. The typical FOVs, image quality, and appearance of the lung structure as well as the amount of data can be inferred, and inspected in the reconstructions provided online (https://doi.org/10.5281/zenodo.3892637). The datasets are denoted by patient I-VI, respectively. Gray values of the tomographic reconstruction represent phase shift per voxel with edge length $vx$, the local electron density difference to the average paraffin can be computed by

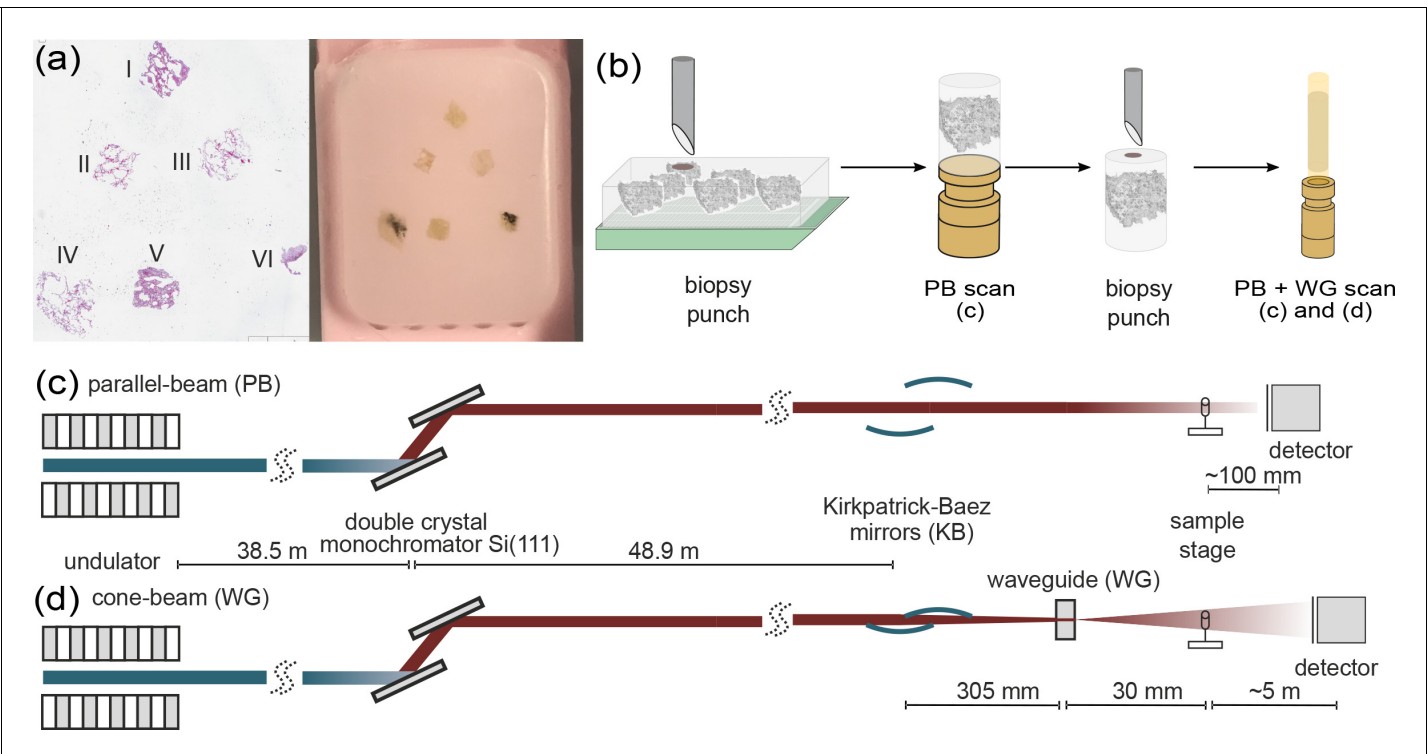

**Figure 1.** Multi-scale x-ray tomography setup. (a) HE stain of a tissue micro-array paraffin block with samples of all six patients who succumbed to Covid-19. (b) Schematic representation of the sample preparation and mounting. In a first step biopsy punches containing the entire individual lung tissues are transferred onto a holder for the parallel-beam local tomography acquisition, followed by a further reduction in size to a 1 mm biopsy punch, for cone-beam tomographic recordings. (c) Configuration for parallel-beam tomography. (d) Configuration for cone-beam holo-tomography.

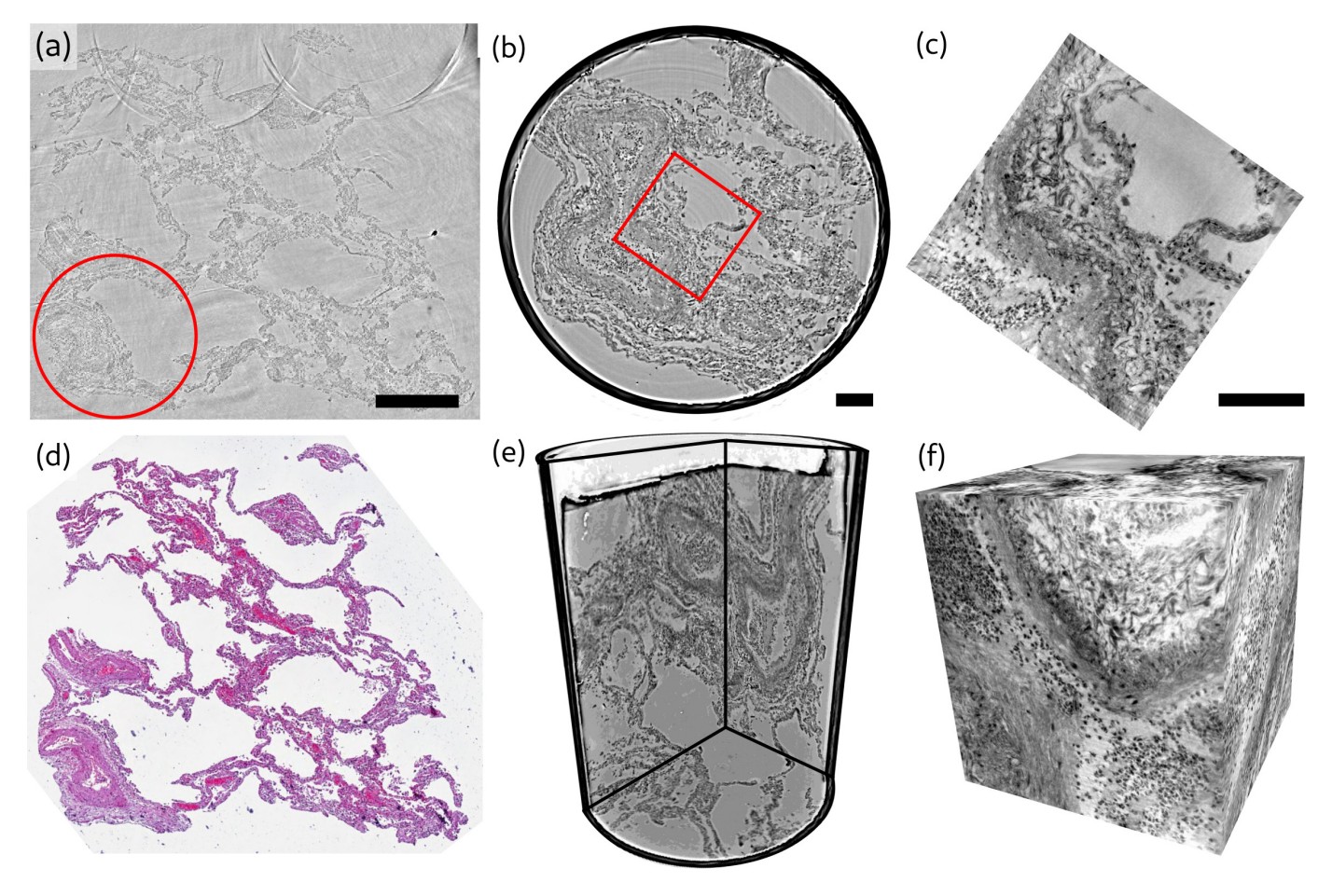

**Figure 2.** Overview of reconstruction volumes, exemplified for a punch biopsy of pulmonary autopsy sample I. (a) Virtual section through the stitched reconstruction volume with indicated position for the 1 mm biopsy punch. (b) Virtual section through the reconstruction of the 1 mm biopsy punch volume with indicated position for the cone-beam dataset. (c) Virtual section through the cone beam dataset. (d) Image of an HE-stained histological section, which is in close proximity to the virtual slice in (a). (e) 3D views of the biopsy punch. (f) 3D views of the cone-beam datasets. Scale bars: (a) $500\,\mu$m and (b,c) $100\,\mu$m. The entire reconstruction volume is visualized in *Video 1*.

$$\Delta\rho_e(\mathbf{r}) = \frac{\varphi_{vx}(\mathbf{r})}{vx \cdot \lambda \cdot r_0}, \tag{1}$$

with wavelength $\lambda$ and $r_0$ classical electron radius.

## Reconstructed electron density

Next, we present representative slices through the reconstruction volumes of all samples for all acquisition scales. *Figure 3* presents the stitched reconstruction volumes, recorded under conditions of local tomography, see Table 3. Conventional HE-stained histology images of all samples are shown in *Appendix 1—figure 1*. Since these volumes are computed from stitching up to 20 individual tomograms, the question arises to which extend the image quality is limited by potential artifacts of local tomography, that is, errors due to the fact that part of the sample is outside the reconstruction volume. For this reason, 1 mm punches were taken after the stitched overview and rescanned in the parallel beam configuration, without local tomography conditions, since they fitted within the FOV. The results are presented in *Figure 4*, and validate the previous stitching results. The 1 mm punches then also provided an appropriate size for the cone-beam recordings, which are shown in *Figure 5*. Importantly, in each scan the previous level guided the choice for the next FOV and informed about the larger environment. In the following, we briefly discuss the samples one-by-one,

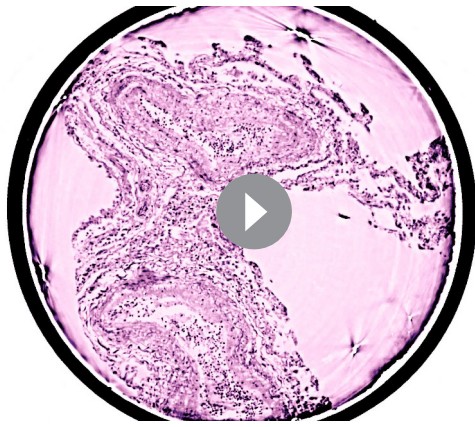

**Video 1.** A video showing a fly-through the reconstruction volume of sample I. First the parallel beam data set is shown, followed by the cone beam

with regard to all acquisition scales. A comparison of morphological features between conventional and virtual histology is shown in the Appendix 1.

## Sample I

By conventional histopathological assessment, the peribronchial alveolar parenchyma of sample I showed DAD with focal formation of hyaline membranes adjacent to the epithelial lining, moderate lymphocytic interstitial pneumonia and singular thrombi in small pulmonary veins. There is a moderate hypertrophy of the muscular media in smaller pre- and post-capillary blood vessels with desquamation of the endothelial cell layer as well as mild centrilobular emphysema (original magnification 100×). In PC-CT, enlarged alveolar

**Figure 3.** Stitched parallel-beam reconstructions for full pulmonary samples (I–VI). Representative virtual sections through the reconstruction volumes of full biopsies (I–VI), respectively. Scale bars: 500 $\mu$m.

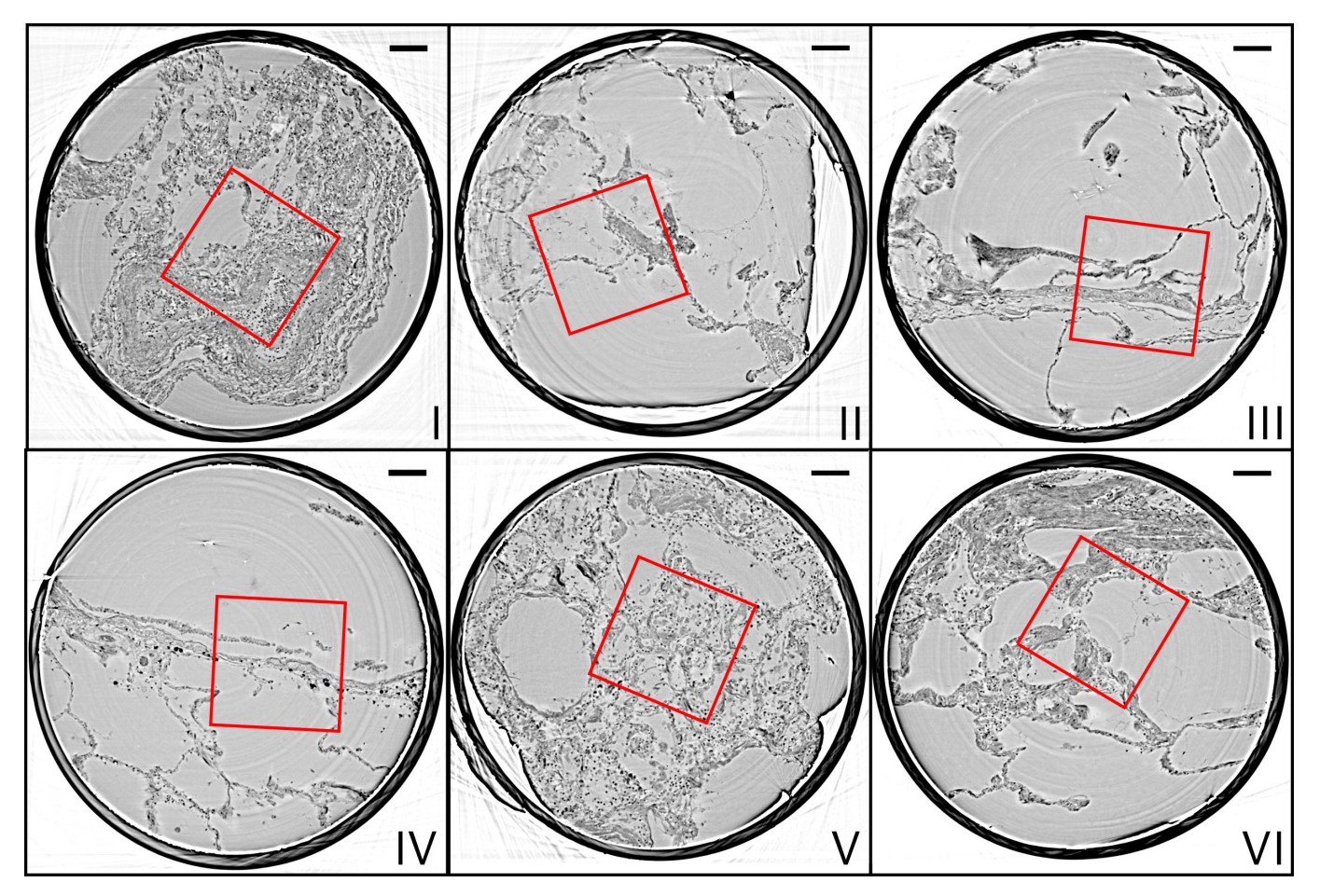

**Figure 4.** Parallel-beam reconstructions for 1 mm biopsy punches (**I–VI**). Representative virtual sections through the reconstruction volumes of the 1 mm punches into the volumes of the full biopsies (**I–VI**), respectively. The fact that the punches are isolated results in higher image quality, since the errors associated with local tomography are avoided. Scale bars: $100\,\mu\text{m}$.

septa with pronounced lymphocytic inflammation are displayed. The reconstruction volume contains a large artery filled with erythrocytes (*Figure 3-I*, lower left), which bifurcates into two vessels. This area was then selected for the 1 mm biopsy punch extraction. The cone-beam zoom tomogram was then centered around the perimeter of the blood vessel. This volume is particularly well suited to investigate the connective tissue including elastic fibers and collagen, as well as smooth muscle.

## Sample II

Histomorphological analysis shows peribronchial alveolar parenchyma with hyperemia of capillary and post-capillary blood vessels, as well as a moderate centrilobular emphysema (original magnification 100×). On the level of blood vessels, both blood-filled and empty vessels are discernible. It should be noted that septa with signs of parallel capillaries are visible. In the reconstruction volume of the zoom tomogram, a single vessel can be easily tracked over large distances.

## Sample III

The sample consists of peribronchial alveolar parenchyma showing prominent multifocal neutrophilic capillaritis as well as a moderate centrilobular emphysema (original magnification 100×). In PC-CT, septa with again similar physiological size and distribution emerge, with moderate emphysema. The bottom part of the sample contains a fibrous area near a larger blood vessel. The zoom tomogram shows a single septum, a blood vessel and a fibrous region.

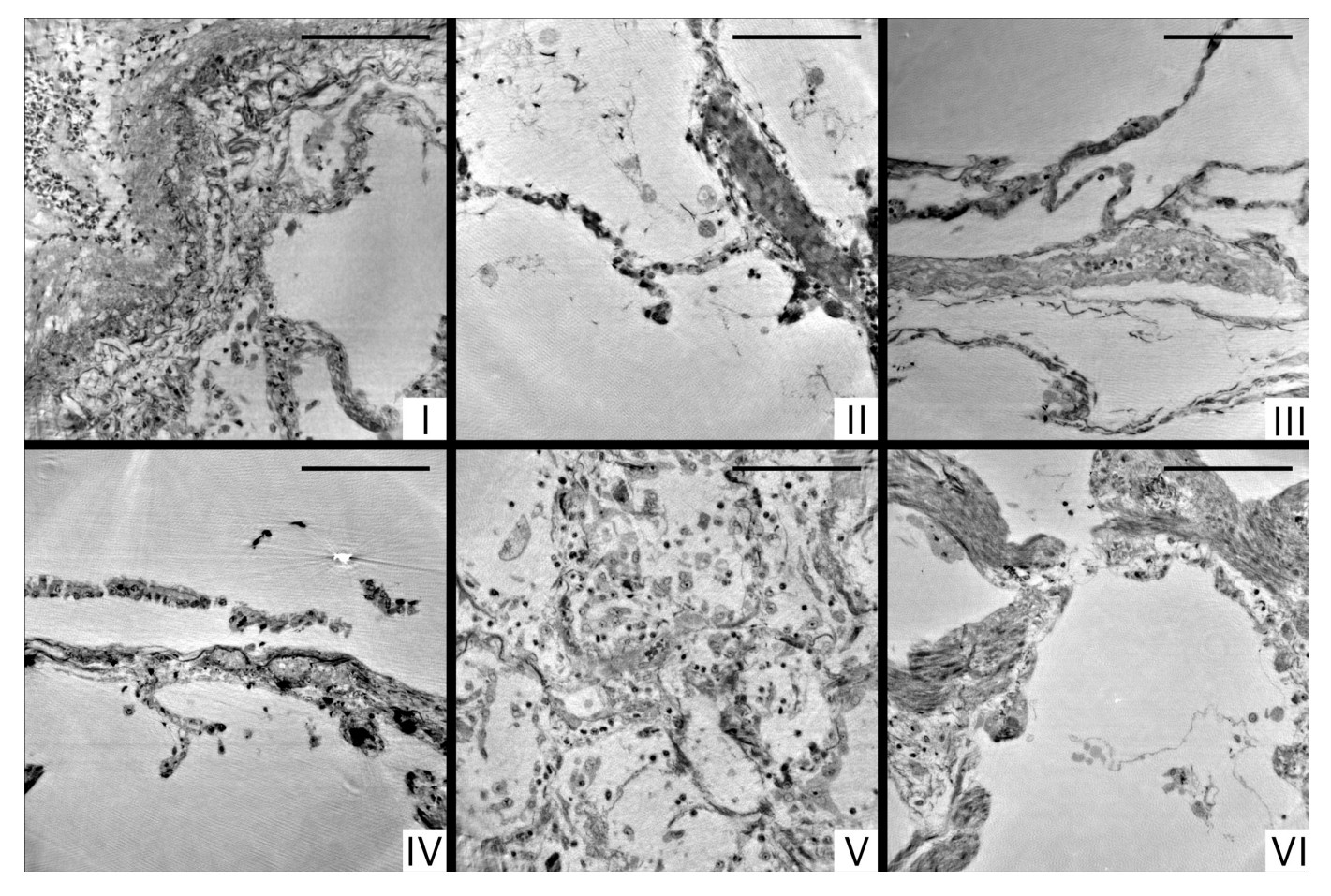

**Figure 5.** Cone-beam reconstructions for biopsy punches (**I–VI**), shown for approximately the same slices as in *Figure 4* for the parallel beam reconstruction. Virtual sections through the reconstruction volumes of the cone-beam recordings corresponding to sections in *Figure 4*, obtained by the parallel beam configuration, for biopsies (**I–VI**) , respectively. Scale bars: $100\,\mu m$.

## Sample IV

Histomorphological analysis shows peribronchial alveolar parenchyma with marked lymphocytic interstitial pneumonia, multifocal venous thrombi and focal intraalveolar fibrin deposition in terms of DAD. Furthermore, there is a mild centrilobular emphysema (original magnification 100×). In PC-CT, a network of thin septa, thrombi and emphysema, as well as a large empty blood vessel appears. Electron-rich diffuse black granules are also visible. The biopsy punch was selected to contain the empty blood vessel, some small thrombi. It also includes thin septa and tissue embedded dirt-particles. The zoom tomogram covered tissue with black granules as well as a band of inflammatory cells next to a septum wall.

## Sample V

By conventional histological assessment, Sample V consists of peribronchial alveolar parenchyma showing massive lymphocytic interstitial pneumonia with ubiquitous hyaline membranes superimposed on the alveolar walls, neutrophilic capillaritis and multifocal post-capillary thrombi in terms of severe DAD. Furthermore, bronchialized alveolar epithelial cells show cytopathic changes and multifocal desquamation, as does alveolar macrophages. Focally, accumulation of intraalveolar neutrophilic granulocytes in the sense of a florid bronchopneumonia can be observed (original magnification 100×). PC-CT data give rise to alterations of the overall morphology due to Covid-19, including substantial inflammation, pronounced hyaline membranes, and high load of lymphocytes.

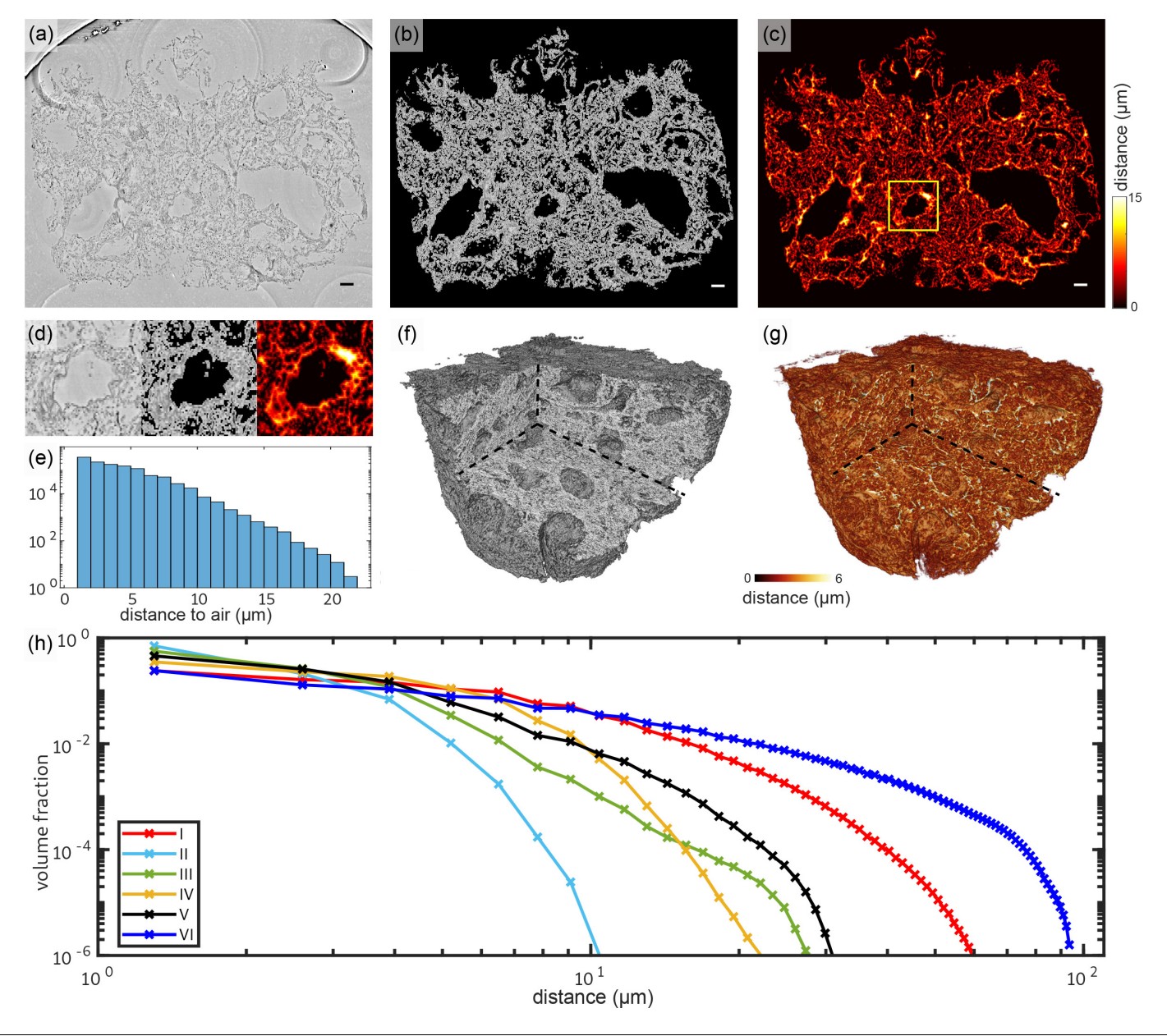

**Figure 6.** Tissue compactness and distance metrics: The 3D reconstructions of the lung tissue were analyzed in terms of specific surface and characteristic lengths. The workflow is sketched for sample V in the top part (**a–g**). In a first step, the tissue was segmented using Ilastik. A slice of the reconstructed electron density is shown in (**a**). Based on the segmentation, the areas of air which are directly connected and potentially filled with oxygen (or blood) are masked out (**b**). (**c**) For each of the remaining tissue voxels the shortest possible distance to air was calculated. Especially around vessels and larger alveoli, the distances are larger. (**d**) Zoom into an area around a vessel. Further analysis is based on the distance distribution shown in (**e**). (**f**) Volume rendering of the reconstructed electron density, with (**g**) showing the corresponding 3D distance map. (**h**) Based on the tissue segmentation of all samples, the distance from the tissue interior to the closest air compartment was calculated. In order to compare all samples, the count of voxels was normalized by the total volume of the respective sample. The specific surface area $S_V$ (represented by the first value of each curve), the characteristic length $L_c$ and the mean distance $\overline{d_{O_2}}$ for each sample was calculated based on this data. Double logarithmic scale, bin width of the distribution of distances: 1 $vx$. Scale bars: (**a–c**) 100 $\mu$m.

The biopsy punch was chosen to include areas with increased presence of hyaline membrane and lymphocytes. A blood vessel splitting into several smaller blood vessels is easily recognized when browsing through the reconstructed volume. Noteworthy, different cell types as macrophages, T-cells or erythrocytes can be distinguished in the zoom tomogram.

**Table 1.** Results of the analysis of tissue characteristics: specific surface area $S_V$, characteristic length $L_c$ and mean distance $\overline{d_{O_2}}$ and standard deviation from all tissue voxels to air as well as the mean concentration of lymphocytes $c_l$ for all six Covid-19 positive samples as well as for one control sample. Colors match the distance graphs in **Figure 6**.

| Patient no. | $S_V$ (%) | $L_c$ (µm) | $\overline{d_{O_2}}$ (µm) | $c_l$ ($10^5/\text{mm}^3$) |
|---|---|---|---|---|
| I | 13.87 | 9.4 | 5.9±5.3 | 16.0 |
| II | 46.56 | 2.8 | 2.1±1.0 | 14.1 |
| III | 33.75 | 3.9 | 2.5±1.5 | 4.4 |
| IV | 25.90 | 5.0 | 3.2±2.3 | 7.1 |
| V | 19.28 | 6.7 | 3.6±2.1 | 4.8 |
| VI | 11.87 | 11.0 | 9.1±10 | 6.1 |
| CTRL (hyd.) | 20.04 | 6.5 | 5.0±5.1 | - |

## Sample VI

Histomorphological analysis shows peribronchial alveolar parenchyma with lymphocytic interstitial pneumonitis and a singular thrombus in a small vein. The interstitium of the alveolar septae is widened by myogenic metaplasia. Adjacent, centrilobular emphysema and anthracosis are observed. The bronchial mucosa shows varying degrees of lymphocytic inflammation in the sense of chronic bronchitis/bronchiolitis (original magnification 100×). From PC-CT reconstructions, the sample consists of thin alveoli (**Figure 3VI**, upper left) evolving into compact, fibrotic tissue (**Figure 3VI**, lower right). The amount of lymphocytes is rather low in this sample. Black granules and some thrombi are embedded within the bulky tissue parts. The biopsy punch covers the region of transition from alveoli to fibrotic tissue, containing also a thrombus and capillaries as identified from the zoom tomogram.

## Visualization of pathologies, segmentation, and quantification

The overview scans of all paraffin embedded Covid-19 positive samples and one biopsy of a hydrated control lung were analyzed in terms of structural characteristics. The top row of **Figure 6** shows the workflow of the analysis on the example of the 3D reconstruction of sample V. Based on the tissue mask the distances for each tissue voxel to the next voxel containing oxygen were determined. For the analysis of the tissue it is mandatory to consider the three-dimensionality of the samples, similar to the analysis of porous structures (**Müller et al., 2002**). More generally, we claim that 3D histology analysis based on x-ray tomography and light sheet microscopy (**Power and Huisken, 2017**) is required to quantitatively understand functional tissue properties based on its 3D architecture. Here, this can also be seen in **Figure 6d**, which shows a zoom of the distance analysis around a small blood vessel which is marked with a yellow box. While the wall thickness appears quite homogeneous in the 2d slice, the distance analysis reveals that the vessel is thicker on the top right.

The distribution of the distances obtained for a given slice is shown in **Figure 6e**. The swelling of the alveolar walls as well as the inflamed blood vessels can be identified by comparing the reconstructed electron density and the 3D-distance map (see **Figure 6f and g**). **Figure 6h** shows the distribution of the tissue-air distances (histogram) for all samples, following the workflow illustrated in **Figure 6e**. The binning of distances was set to one voxel length. The figure underlines the high diversity of the tissue structure which could already be seen in the 3D histology. Further, it directly informs about the specific surface $S_V$, which is given by the first point of the graph. The corresponding parameters and metrics are tabulated in **Table 1** for all samples. Additionally, the mean concentration of lymphocytes $c_l$ within the lung tissue is listed for all samples. The values quantify the general structure of the tissue which is qualitatively discernible by eye. Samples with a high amount of swollen, inflamed blood vessels and thick hyaline membranes exhibit a larger characteristic length. Note, that the control lung was prepared in a hydrated environment and shrinking due to further preparation of the sample does not occur. Hence, the results cannot be directly compared to the paraffin embedded samples. Further, the analysis of the lymphocyte concentration was performed since no lymphocytes were found in the reconstructed volume. The low values of $L_c$ and $\overline{d_{O_2}}$ for

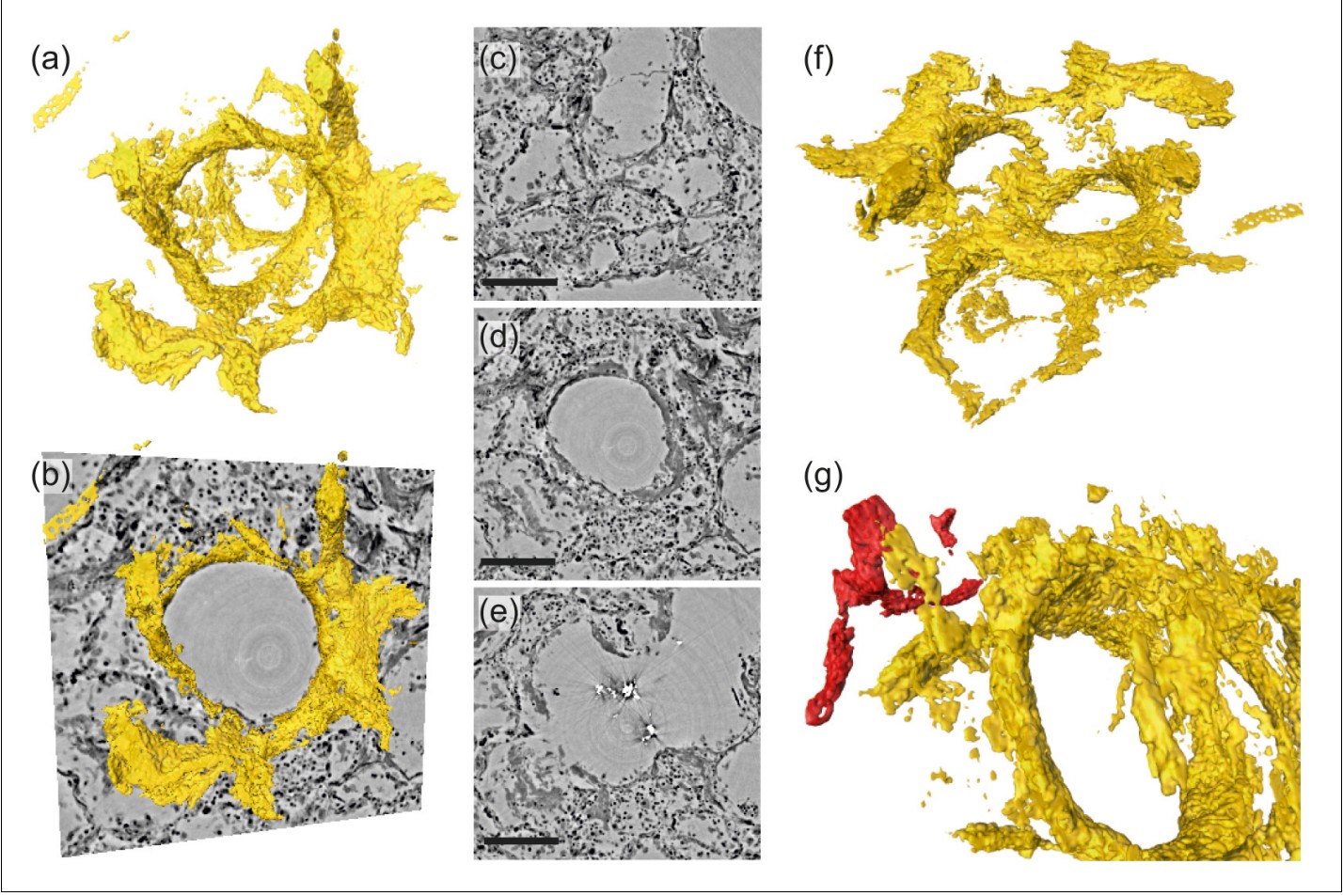

**Figure 7.** Rendering of hyaline membrane attached to alveolar walls (patient V, parallel beam-scan of a 1 mm punch). The rendered subvolume was restricted to $1.15 \times 1.10 \times 0.56 \ \mathrm{mm}^3$, to contain a single alveole foremost. (a) Volume rendering of the segmented hyaline membrane in same spatial orientation as (c)-(e), which show virtual slices through the (c) top, (d) center and (e) bottom of the alveole. For a better spatial classification, (b) gives a combination of the volume in (a) and the slice in (d). (f) Volume rendering of the entire subvolume including neighboring alveolae. (g) Zoom-in onto a major blood vessel (red) which is directly connected with the hyaline membrane. Scale bars: (c–e) 300 $\mu$m.

sample II correlate with the lack of ground-glass opacification in clinical CT. Based on the extracted structural parameters, the degree of inflammation and swelling of lung tissue can be evaluated. E.g. patient II has the highest surface area volume-ratios while sample I and VI have a relatively low specific surface. Larger characteristic lengths may also be indicative of inflammation and the formation of hyaline membranes, which will be evaluated in the following based on ROI and high-resolution reconstructions.

*Figure 7* illustrates the aggregation of hyaline membrane in the vicinity of a single alveole. Volumetric renderings in *Figure 7a and b* demonstrate particular attachment of fibrin to the alveolar walls. In cases of severe hyaline membrane formation as for this patient, this pathological alteration can be tracked throughout the volume *Figure 7c–e*. In *Figure 7f*, hyaline membranes of neighboring alveoles are indicated. In the 3D-context, their locations with respect to blood vessels can be inspected, see *Figure 7g*, which exemplifies a direct connection of hyaline membranes to the vasculature.

The severeness of hyaline membrane formation is case-specific, as the yellow rendering in *Figure 8a,b,d* demonstrates reduced amounts of deposits for patient I in a subvolume of parallel-beam reconstructions. Further, lymphocytes (red) were identified based on the automated cell segmentation (see Materials and methods). For clearer visualization, each cell is rendered as a sphere with a size corresponding to the mean cell volume in *Figure 8a,b,d*. Based on convolution of the cell

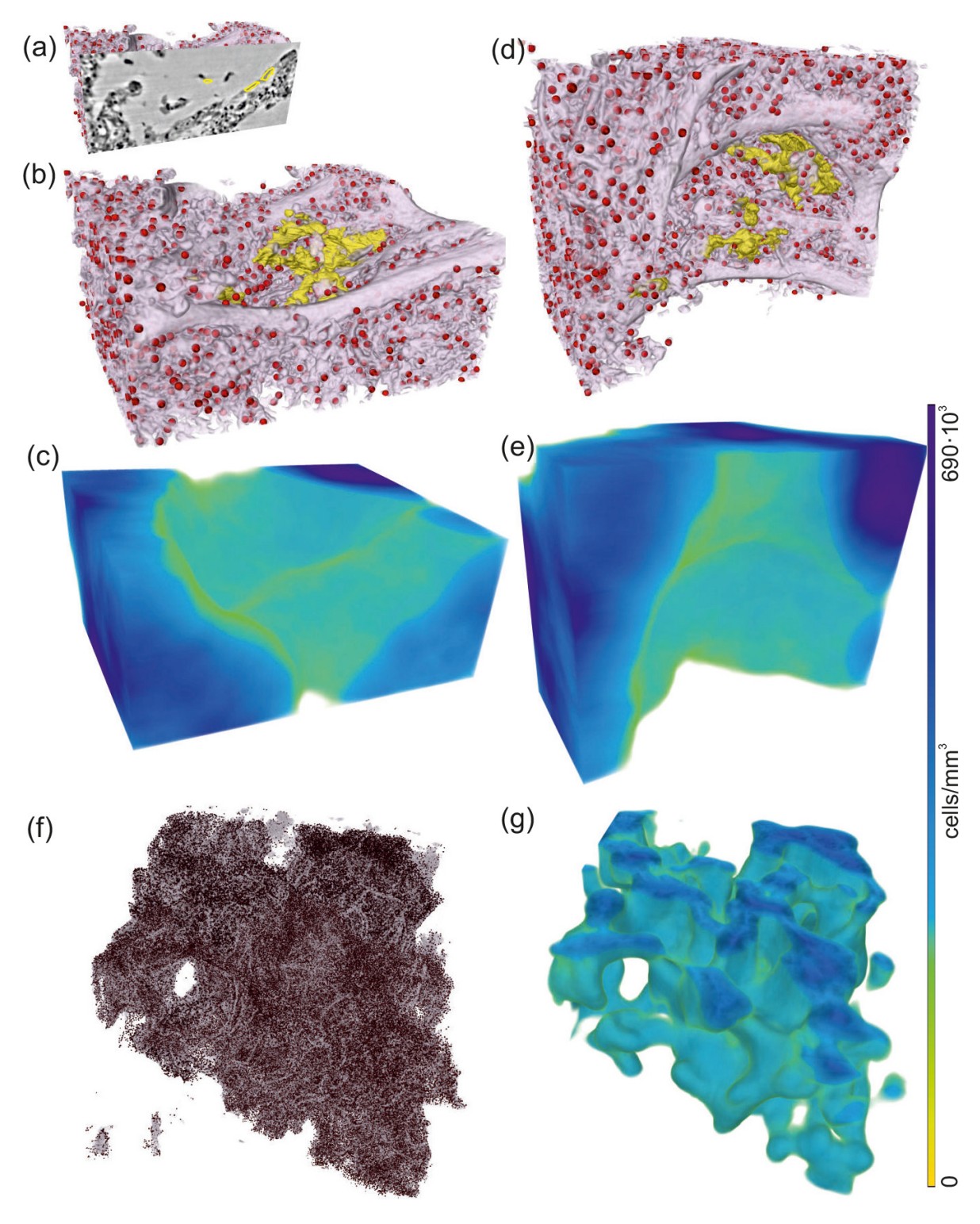

**Figure 8.** Rendering of alveolar wall with hyaline membrane and quantification of lymphocyte infiltration (unstained tissue, patient I, parallel beam scan of a 1 mm punch). The illustrations in (a–e) show a subvolume of $0.60 \times 0.48 \times 0.26$ mm$^3$, (f and g) this concept applied to the full tissue block of $2.57 \times 2.99 \times 0.98$ mm$^3$. (a) Yellow contours mark the locations of hyaline membrane in an exemplary slice. In same spatial orientation, (b, d and f) volume rendering of soft tissue (light pink), infiltrated by hyaline membrane (yellow) and lymphocytes (red) and (c, e and g) their local cell density among the lung tissue, including air compartments.

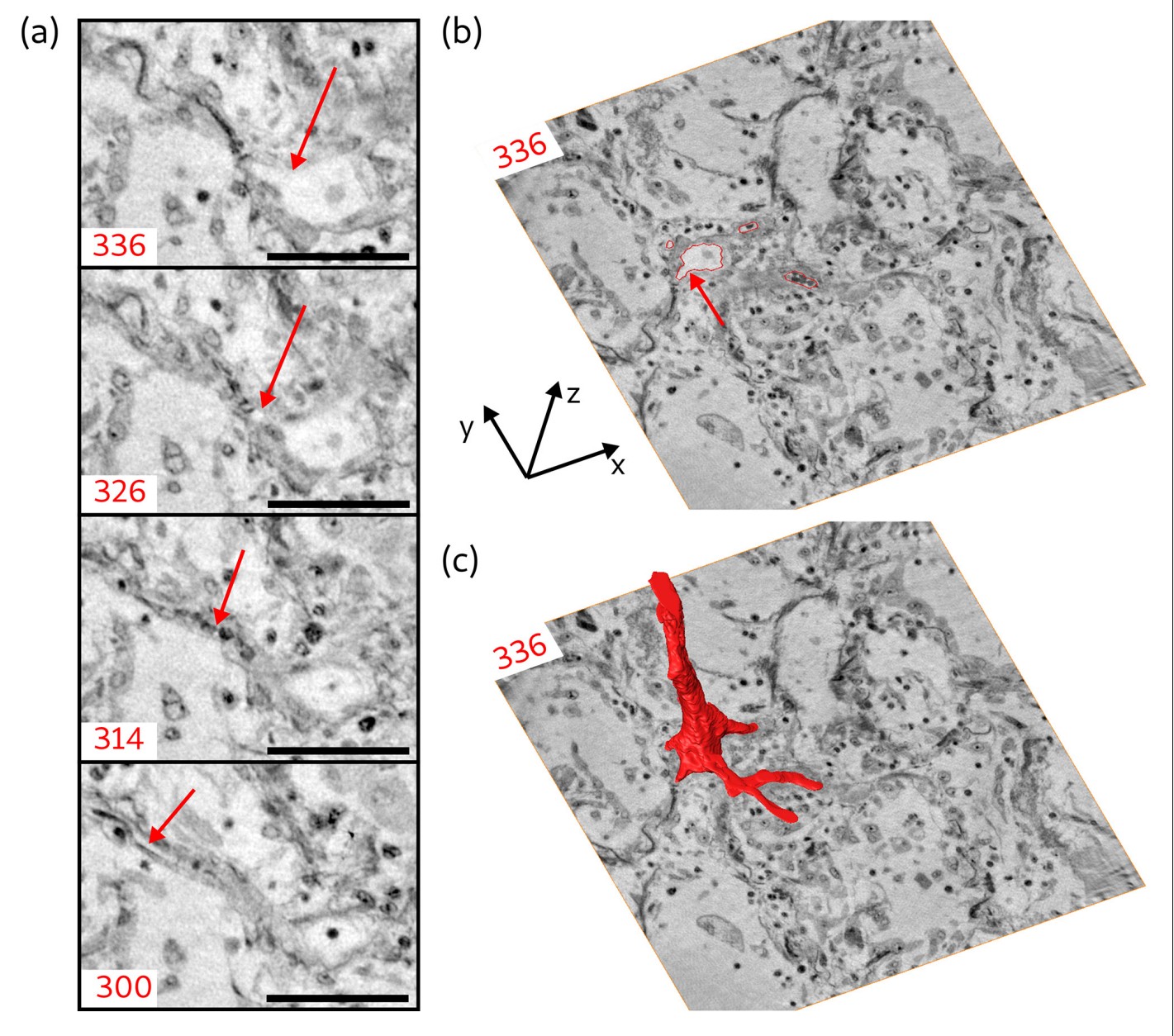

**Figure 9.** Segmentation of the blood vessel network, exemplified for biopsy (**V**). (**a**) Series of slices in z-direction illustrate the separation of a capillary ($\approx 5\,\mu\mathrm{m}$ diameter) from a blood vessel. Red arrow marks the separating capillary. In slice 314, the capillary is entirely filled with erythrocytes. (**b**) Red lines in the entire virtual slice (336) represent the contours of a manual segmentation of the blood vessel network. Three capillaries already separated from the main vessel, while the fourth starts to emerge, indicated by the red arrow. (**c**) Segmentation illustrating the 3D structure of the blood vessel network. Scale bars: $50\,\mu\mathrm{m}$.

positions with a sphere of $100\,\mu\mathrm{m}$ in radius, the local cell density was calculated (***Töpperwien et al., 2018***) and presented as 3D-maps of cells/mm$^3$ in ***Figure 8c,e***. This concept was then translated to a 3D-stitched volume of an entire tissue block as shown in ***Figure 8f,g***.

Next, the segmentation of blood vessels is demonstrated for the example of a splitting blood vessel in the zoom tomogram of sample V. The segmentation was performed manually. To give an impression of the separation of a single capillary, a series of virtual slices in the xy-plane (magnified views) is shown in ***Figure 9a***. The separation starts with the creation of a branch from the blood vessel (arrow in slice 336). In slice 326, $1.7\,\mu\mathrm{m}$ above slice 336, this branch evolves into an empty and separated capillary. Another $2\,\mu\mathrm{m}$ above, the capillary is entirely filled with cells. Further $2.3\,\mu\mathrm{m}$, the

capillary is empty again and has a diameter of about $2.8\,\mu\mathrm{m}$. The segmentation of the blood vessel with all its separated branches is indicated for slice 336 in *Figure 9b* by the red lines. In this slice, three capillaries have already separated from the main vessel, while the fourth starts to emerge, indicated by the red arrow. The 3D shape of the blood vessel is illustrated by the 3D rendering of the segmentation, shown in *Figure 9c*. This segmentation of the blood vessel shows the potential of the datasets, which may be fully exploited in future with more advanced segmentations.

## Discussion

In summary, we have demonstrated that multi-scale x-ray phase contrast tomography can firstly augment pathohistology of the lung to full three dimensions, and secondly provide a link between microscopic and macroscopic scales. For the first time, characteristic morphological changes associated with DAD such as hyaline membranes and pronounced inflammation have been imaged in three dimensions, in particular in Covid-19, the most severe pandemic which mankind has faced for decades. Moreover, we conducted our study in FFPE material, a fixation method established for well over a century and ubiquitously available.

The advantage of 3D histology compared to conventional histology relates to the simple fact that features can be visualized and investigated in their full native dimensionality. This enables the accurate determination of cell densities, the quantification of morphological parameters such as the distance metrics shown in *Figure 6*, and tracing of blood vessels. Standard histology and in particular immunohistochemical staining, on the other hand, can inform us more precisely on the biomolecular origin of the structure, while immunostaining coupled with x-ray contrast agents is not yet within reach. Importantly, since PC-CT is nondestructive, histology can be performed on well-chosen sections after the scan, see Appendix 1.

On the side of imaging technology and exploitation, there is still ample room for improvement: higher resolution could be achieved by further geometric zooms, waveguide optics with higher numerical aperture, in combination with pixel detector technology, and further improvements in holographic reconstruction. Equally important to extend the length scales covered to small scales is the further upscaling of the FOV based on stitching different sub-tomograms. At the same time the scales should be extended by including more than two zoom levels, based on different detector and illumination settings. In this way, one could bridge scales to cover the entire lobe of a lung. To this end, optimized recording and data flow is a larger bottleneck than photons and optics, in particular since 4th generation synchrotron radiation sources are about to deliver unprecedented brilliance. Finally, in view of exploitation and information gain, specific labels for different cell types and 3D immunostaining coupled to radiocontrast agents should be developed. Here, lung offers an advantage over other tissue in view of volume accessibility and label diffusivity.

Most importantly, efforts have to be directed to exploit the full information contained in the reconstruction volumes based on advanced segmentation. Tracing of capillaries is an obvious important next step, in view of unraveling the role of intussusceptive angiogenesis in Covid-19 (*Ackermann et al., 2020*). The current data, which are made fully available at https://doi.org/10.5281/zenodo.3892637, are likely to already provide such clues once that more advanced segmentation approaches based on machine learning are applied. More generally, it may advance our understanding of DAD in the particular case of Covid-19, as an intrinsically volumetric phenomenon. Beyond the current data, further studies could shed further light on the differences between moderate and severe progression. Possibly, phase contrast tomography of lung biopsies could in future also help diagnosis and treatment.

## Materials and methods

### Autopsy, clinical background and tissue preparation

In total we investigated six postmortem lung samples from Covid-19 patients (*Menter et al., 2020*). A tissue micro-array paraffin block with samples of all six patients and the corresponding HE stain is shown in *Figure 1a*. Information about age, gender, hospitalization, clinical, radiological and histological characteristics of all patients are shown in *Table 2*. All patients suffered from hypertension and were treated with RAAS (renin-angiotensin-aldosterone-system) interacting drugs.

**Table 2.** Sample and medical information.

Age and gender, clinical presentation with hospitalization and treatment; NIV: non-invasive ventilation, I: immunosuppression, S: smoker, GGO: ground-glass opacification, C: consolidation, DAD: diffuse alveolar damage.

| Sample no. | Age group, gender | Hospitalization, clinical, radiological, and histological characteristics |
|---|---|---|
| I | 60–70, F | 5-10d, GGO, DAD |
| II | 80–90, M | 5-10d, C, I, DAD |
| III | 90–100, M | 1-4d, GGO, C, DAD |
| IV | 70–80, M | 1-4d, NIV, S, GGO, C, DAD |
| V | 60–70, M | 5-10d, NIV, S, GGO, C, DAD |
| VI | 70–80, M | 1-4d, S, GGO, C, DAD |
| CTRL | 20–30, M | - |

Heterogeneous ground glass and consolidation were observed in all patients clinical CT scans and the cause of death was also related to respiratory failure in each patient (patient IV cardio-respiratory failure). Additionally, tumor-free lung samples from partial resections of pulmonary carcinomas were analyzed as a reference.

From each of the six Covid-19 patients, two tissue samples with edge lengths of about 4 mm each were analyzed. To one sample of each patient, a metal containing stain (uranium acetate, UA) was applied, the other samples remained unstained. Separated for their stain, six tissue samples were dehydrated and embedded in the same multi-sample paraffin block. The size of the postmortem tissue samples made available for the study varied between the different patients (I-VI), with maximum cross-section of about 4 mm after dehydration. From all six samples, biopsy punches were taken by either a 8 mm or a 3.5 mm punch, depending on the individual size. The punches were then transferred onto a holder for the parallel-beam local tomography acquisition, followed by a further reduction in size (after measurement of the entire sample) to a 1 mm biopsy punch, for further tomographic recordings. A sketch of the sample preparation is shown in *Figure 1b*. The control lung sample was first mounted in an Eppendorf tube for parallel beam acquisitions, and a 1 mm biopsy punch was then transferred into a polyimide tube similar to the paraffin-embedded ones, but scanned in fixative buffer solution.

## Phase contrast tomography

For Covid-19 lung tissue, the scans were recorded at the GINIX endstation of the PETRA III storage ring (DESY, Hamburg). The projections were acquired at two different photon energies $E_{ph}$, $8\,keV$ and $13.8\,keV$, using the first and third harmonic of the 5 m P10 undulator and a Si(111) channel-cut monochromator, respectively. Data are shown here only for $8\,keV$, which gave highest contrast for the unstained lung tissue. Two tomography configurations were combined to cover a larger range of length scales:

### (1–*parallel beam*)

Recordings with the unfocused quasi-parallel beam illumination and a high resolution microscope detection system, resulting in in a FOV of 1.6 mm sampled at a pixel size of 650 nm.

### (2–*cone beam*)

Holographic recordings with the divergent and coherence filtered beam emanating from a compound focusing system composed of a Kirkpatrick-Baez (KB) mirrors system and an x-ray waveguide, resulting in a FOV of 0.4 mm sampled at a pixel size of about 167 nm (depending on exact geometry).

Both configurations were implemented side-by-side, using the same fully-motorized tomography stage and mounting, as detailed in *Frohn et al., 2020*. A sketch of both configurations is shown in *Figure 1* . First, parallel-beam overview scans were acquired of the entire tissue volume embedded in paraffin. To this end a 3.5 mm biopsy punch was taken from the multi-sample tissue block, and then scanned in a stitching-mode yielding a large overview reconstruction, composed of up to 20

individual tomograms (depending on tissue size). To increase image quality and avoid artifacts related to region-of-interest (ROI) or *local* tomography, a selected 1 mm punch was then taken from the already scanned larger tissue cylinder, and re-scanned, first in the parallel- and then the cone-beam geometry. Experimental and acquisition parameters used for the data shown are listed in *Table 3*.

The configuration for (1–*parallel beam*) is depicted in *Figure 1c*. The high-resolution microscope detection system (Optique Peter, France) was based on a $50\,\mu m$ thick LuAG:Ce scintillator imaged with a 10× magnifying microscope objective onto a sCMOS sensor (pco.edge 5.5, PCO, Germany), resulting in an effective pixel size of $0.65\,\mu m$. The high photon flux density allowed for image acquisition with continuous motor movement, short acquisition time of 35 ms per frame, and a framerate of 20 fps. Single-distance tomogram recordings with about 1500 projections, and flat images before and after the scan, took less than 2 min. For these scans, the focusing optics (KB-mirrors and waveguide) as well as the fastshutter (Cedrat technologies) were moved out of the beam, and beam size was adjusted by the upstream slit systems. To avoid detector saturation, the beam was attenuated by 4× single crystal silicon wafers.

The configuration for (2–*cone beam*) is depicted in *Figure 1d*: The beam was focused by the KB-mirrors to about 300 nm. To further reduce the secondary source size, to increase coherence, and to achieve a smooth wavefront for holographic illumination, an x-ray waveguide formed by 1 mm long lithographic channels in silicon with a cross-section of about 100 nm was positioned in the focal plane of the KB-mirrors, resulting in an exit flux of $1\text{–}4\cdot10^9\,\mathrm{photon/s}$ (depending on alignment and storage ring), as measured with the single photon counting detector (Pilatus, Dectris). The sample was positioned at variable (defocus) distances behind the focus (waveguide exit), typically $z_{01} = 125\,\mathrm{mm}$ for the first distance. The geometrically magnified holograms were recorded by a fibre-coupled sCMOS sensor (Zyla HF 5.5 detector, Andor Technologies) with a customized $15\,\mu m$ thick Gadox scintillator, and $6.5\,\mu m$ pixel size. The detector position at about $z_{02} = 5100\,\mathrm{mm}$ behind the focus resulted in a magnification of about $M = 41$ (for the first defocus distance), and hence an effective pixel size of 167 nm. 1442 projections were recorded, with a typical exposure time of 2 s per projection.

## Phase retrieval, image reconstruction, and segmentation

### Phase retrieval and reconstruction

Phase retrieval was performed from dark and empty beam corrected holograms, using both linearised single step CTF-approach (*Cloetens et al., 1999*; *Turner et al., 2004*), and non-linear generalizations of the CTF-method based on Tikhonov regularization (NL-CTF), using our code package HoloTomoToolbox as described and deposited (*Lohse et al., 2020*). Importantly, both CTF and NL-CTF implementations can be augmented by imposing support and range constraints, when needed. When available, projections recorded at several defocus distances were first aligned with sub-pixel accuracy and then used for multi-distance phase retrieval. For these purposes, the HoloTomoToolbox provides auxiliary functions, which also help to refine the Fresnel number or to correct for drift in the illumination function. After phase reconstruction of all projections, the tomographic

**Table 3.** Data acquisition parameters for x-ray phase-contrast tomography measurements.

|  | Cone geometry | Parallel geometry |
| --- | --- | --- |
| FOV | 0.4 mm × 0.35 mm | 1.6 mm × 1.4 mm |
| Pixel size | 167 nm | 650 nm |
| $z_{01}$ | 125 mm | - |
| $z_{12}$ | 4975 mm | 10 mm − 100 mm |
| Regime | holographic | direct contrast |
| Rotation | start-stop | continuous |
| Exposure | 2 s | 0.035 s |
| Total exposure | $\simeq 63\,\mathrm{min}$ | $\simeq 75\,\mathrm{s}$ |
| Volumetric flowrate | $1.16 \times 10^4\,\mu m^3/s$ | $3.75 \times 10^7\,\mu m^3/s$ |

reconstruction was carried out with the MATLAB implemented iradon-function (Ram-Lak filter) for the parallel geometry and with the FDK-function of the ASTRA toolbox (*van Aarle et al., 2015*; *van Aarle et al., 2016*) for the cone beam geometry. Hot pixel and detector sensitivity variations as well as strong phase features in the parallel beam illumination resulting from upstream window materials, which persist after empty beam correction, can all result in ring artifacts in the tomographic reconstruction, in particular as these flaws can increase by phase retrieval. To correct for this, the extra information provided by 360˚ scans was used to mask out the corresponding pixels and replace them with values of the opposing projection. Stitching of reconstructions from different tomographic scans was performed using (*Miettinen et al., 2019*). Resolution estimates were obtained by FSC analysis, see Appendix 2.

## Image segmentation and quantification

For each patient the stitched overview scans were analyzed with regard to structural characteristics. The 3D-reconstructions were first binned (2 × 2 × 2), and the tissue was then segmented from the surrounding paraffin using the segmentation software Ilastik (*Berg et al., 2019*), which was then further refined with MATLAB. In order to exclude single macrophages or detached tissue, only voxels connected to the tissue block were considered for the distance map. Further, the areas of the paraffin which represent air compartments were linked to the outside of the tissue block. Individual self-contained areas of paraffin (not connected to air) were excluded from the mask. Based on this segmentation, the distance to the nearest voxel containing oxygen was calculated for each tissue voxel. The *tissue volume V* is given by the sum of all voxels containing tissue. The *surface area SA* is defined by all tissue voxels with a distance of 1 pixel to the air. From this information we calculated the *specific surface*

$$S_V = SA/V \qquad (2)$$

with $SA/V$ surface area volume -ratio. Further a *characteristic length*

$$L_c = V/SA \cdot vx \qquad (3)$$

with $vx$ edge length of a voxel was determined for each sample. Additionally, the *mean distance* $(\overline{d_{O_2}})$ from all tissue voxels to the closest air compartment and its standard deviation was calculated.

Hyaline membranes and capillary networks were extracted using semi-manual segmentation functions in Avizo (Thermo Fisher Scientific, USA). Beside hyaline membranes and the capillary network, different cell types can be readily identified based on the 3D reconstructions of the electron density. In particular, inflammatory cell subpopulations - that is, *macrophages* and *lymphocytes* - can be distinguished. To this end, an automatic and parameter-controlled algorithm denoted as BlobFinder was used (arivis AG, Germany). Based on its segmentation output, the amount and position of the lymphocytes in the 3D-reconstructions from parallel beam scans was calculated. The algorithm is able to identify roundish structures with a given size. For the segmentation of the lymphocytes, we chose a characteristic size of $6.5\,\mu\text{m}$. The structures identified in this step also include macrophages and parts of the capillary system. For the unbinned datasets, a distinction between lymphocytes and the nuclei of the macrophages was made based on the difference in electron density. In the tomographic reconstructions, the lymphocytes appear denser compared to the nuclei of the macrophages. Further, nuclei from endothelial cells and parts of the capillaries filled with blood residues were excluded based on their elongated shape. Only structures with a sphericity higher than 0.55 were included. Based on the segmentation of lymphocytes, the total number of lymphocytes $N_l$ was obtained and the mean concentration of lymphocytes within the lung tissue $c_l = N_l/V$ was estimated.

## Acknowledgements

We thank Maximilian Ackermann and Florian Länger for their helpful suggestions, Patrick Zardo for providing control specimen, Emily Brouwer for help in sample preparation, Bastian Hartmann and Jan Goemann for technical help with instrumentation and IT, and Jakob Koch for help in segmentation. It is also our pleasure to acknowledge DESY photon science management for the Covid-19 beamtime call and beamtime.

## Additional information

### Funding

| Funder | Grant reference number | Author |
|--------|------------------------|--------|
| Bundesministerium für Bildung und Forschung | 05K19MG2 | Tim Salditt |
| H2020 European Research Council | 771883 | Danny Jonigk |
| Max-Planck Schools | Matter to Life | Marius Reichardt Tim Salditt |
| Deutsche Forschungsge-meinschaft | EXC 2067/1-390729940 | Tim Salditt |
| Botnar Research Center for Child Health | BRCCH | Alexandar Tzankov |

The funders had no role in study design, data collection and interpretation, or the decision to submit the work for publication.

### Author contributions

Marina Eckermann, Jasper Frohn, Marius Reichardt, Data curation, Software, Formal analysis, Investigation, Visualization, Writing - original draft, Writing - review and editing; Markus Osterhoff, Software, Investigation, Methodology, Writing - review and editing; Michael Sprung, Fabian Westermeier, Resources, Methodology; Alexandar Tzankov, Resources, Validation; Christopher Werlein, Validation, Investigation, Visualisation; Mark Kühnel, Resources, Validation, Writing - original draft; Danny Jonigk, Conceptualization, Resources, Supervision, Funding acquisition, Validation, Writing - original draft; Tim Salditt, Conceptualization, Resources, Data curation, Formal analysis, Supervision, Funding acquisition, Investigation, Methodology, Writing - original draft, Project administration, Writing - review and editing

### Author ORCIDs

Tim Salditt (iD) https://orcid.org/0000-0003-4636-0813

### Ethics

Human subjects: The study was approved by and conducted according to requirements of the ethics committees at the Hannover Medical School (vote Nr. 9022 BO K 2020).

### Decision letter and Author response

Decision letter https://doi.org/10.7554/eLife.60408.sa1
Author response https://doi.org/10.7554/eLife.60408.sa2

## Additional files

### Supplementary files

• Transparent reporting form

### Data availability

All datasets were uploaded to zenodo: https://doi.org/10.5281/zenodo.3892637.

The following dataset was generated:

| Author(s) | Year | Dataset title | Dataset URL | Database and Identifier |
|-----------|------|---------------|-------------|-------------------------|
| Salditt, T, Frohn J, Eckermann M, Reichardt M, Os- | 2020 | 3d Virtual Patho-Histology of Lung Tissue from Covid19 Patients based on Phase Contrast X-ray | https://doi.org/10.5281/zenodo.3892637 | Zenodo, 10.5281/zenodo.3892637 |

terhoff M, Wester-
meier F, Sprung M,
Tzankov A, Kühnel
M, Jonigk D

Tomography

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

## Appendix 1

### Medical information and correlative histology

Before recording tomographic scans, tissue sections of $2.5\,\mu\mathrm{m}$ thickness were cut from the top, stained by HE (hematoxylin and eosin) and imaged with a microscope. *Appendix 1—figure 1* shows the histological slice of each sample. The imaged section is just above the upper plane of the 3D PC-CT reconstruction volume, but not part of it.

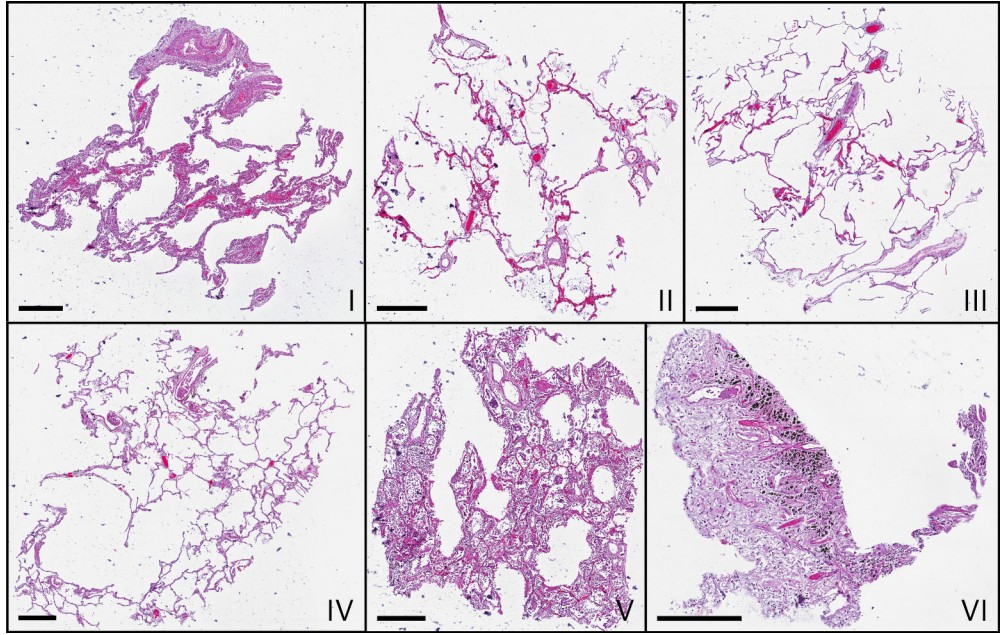

**Appendix 1—figure 1.** Microscopic images of HE-stained histological sections of all samples (**I–VI**). Histological slices show comparable morphologies to the virtual slices in *Figure 3*, which represent different z-position. Scale bars: $400\,\mu\mathrm{m}$.

An overview of different morphological features identified by conventional HE histology and virtual 3D histology is presented in *Appendix 1—figure 2*, while *Appendix 1—figure 3* presents a direct comparison for the same slice. For this purpose, the sample was sectioned and stained after the PC-CT scan. Artery lumen, artery wall, erythrocytes, thrombus, alveolar septum, marcophage, hyaline membrane and black granules (anthracosis) are shown in *Appendix 1—figure 2* for both imaging methods. Contrast of the hyaline membrane is homogenous in both modalities, facilitating idetification and segmentation. Erythrocytes are easily recognized by eye in the conventional histology image, due to the HE staining, but less well distinguished by virtual histology. This results in a difficult differentiation between thrombus and blood stasis as well as a difficult identification of blood capillaries in the alveolar septum. Importantly, however, feature identification can be confirmed by correlative 2d and 3D histology on the same section, as exemplified by *Appendix 1—figure 3*.

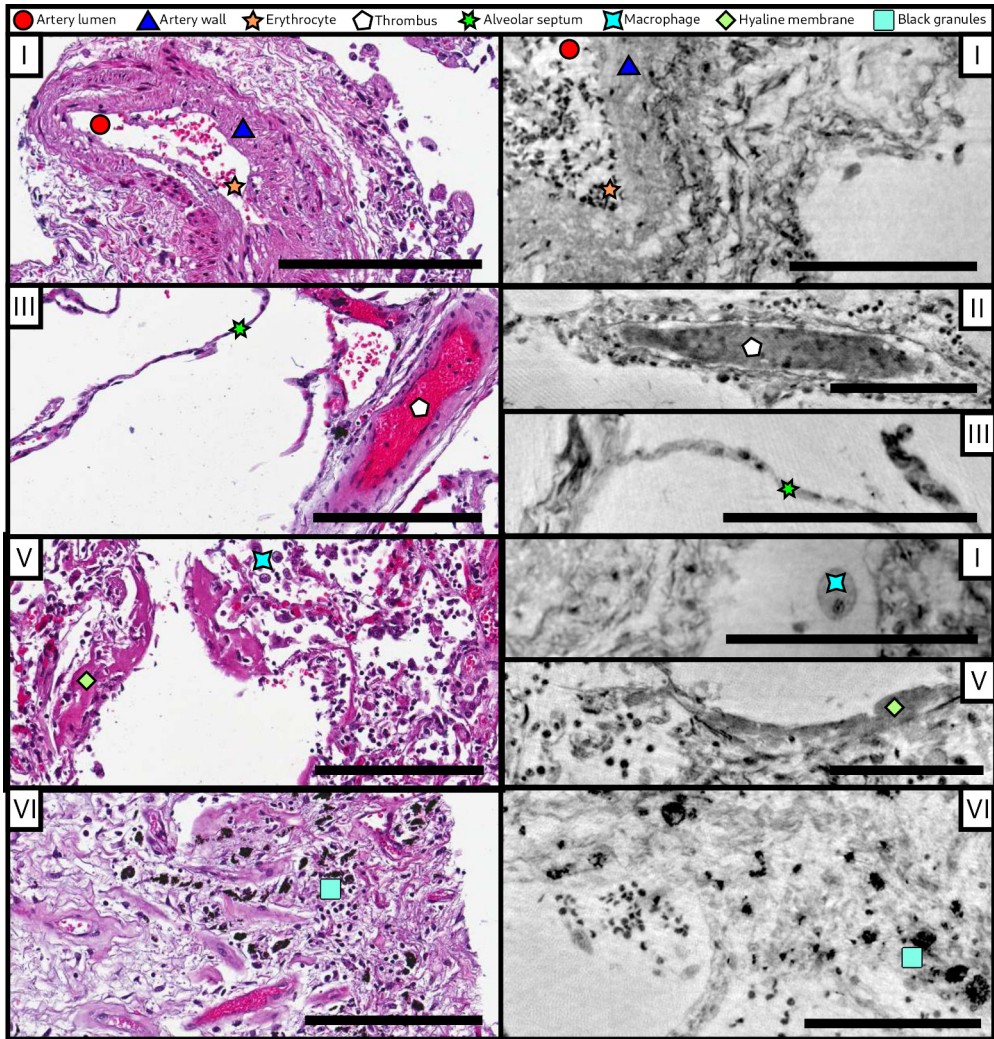

**Appendix 1—figure 2.** Comparison of morphological features between conventional HE histology and virtual histology. Artery lumen, artery wall, erythrocytes, thrombus, alveolar septum, macrophage, hyaline membrane and anthracotic pigments (i.e. the black granules) are presented on exemplary slices of different samples (**I–VI**) for conventional (left) and virtual histology (right). Scale bars: left $200\,\mu$m, right $100\,\mu$m.

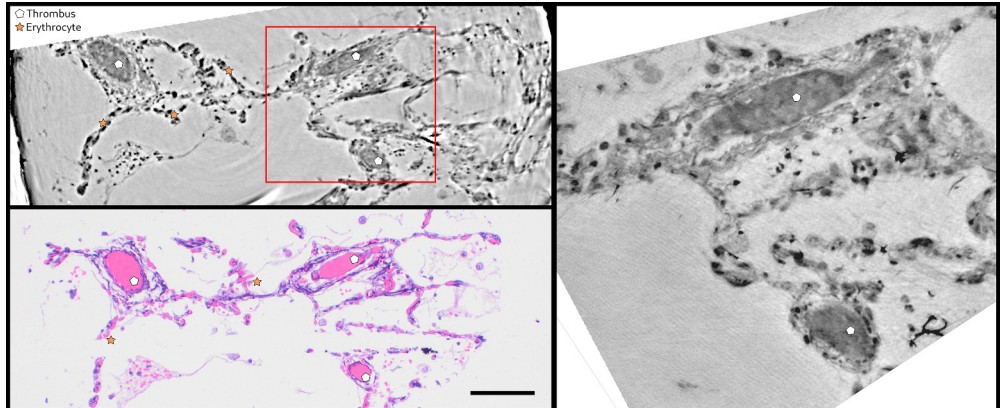

**Appendix 1—figure 3.** Direct comparison of virtual and HE histology for an identical slice/section. (top left) Region of interest of the parallel beam tomogram (Sample II). (bottom left) Corresponding HE stained histology slice. Thrombi and erythrocytes can be identified in both imaging modalities.

The red square marks the position of the zoom tomogram, for which the corresponding slice is shown on the right. Scale bars: $50\,\mu\mathrm{m}$.

# Appendix 2

## Resolution

Resolution estimates are challenging for tissue reconstructions due to the absence of sharp edges or features of well defined size. Here, we follow the approach known as *Fourier-shell correlation* (FSC). Accordingly, an upper bound for the resolution is be obtained in the following way: the CT scan is split into two, and from each half a 3D volume is reconstructed. After registry, that is, mutual alignment of the volumes, the correlation between the two independent reconstructions in Fourier space is plotted as function of spatial frequency. This correlation must not necessarily reflect the pure system resolution, but instead the range of spatial frequencies over which the results are reliable. As such it is not only affected by the system resolution but also by the sample contrast and the noise of the specific scan. Further, it represents an average over all structures, not taking into account that features with stronger/weaker contrast can correspondingly show higher/lower resolution. Here the Fourier operation for FSC was implemented with a Kaiser-Bessel window of 7 pixels. For the parallel-beam data, a central volume of $650 \times 650 \times 650$ voxels was correlated, for the cone-beam data a volume of $685 \times 680 \times 250$ voxels. These sub-volumes were selected to obtain the average values for the tissue while minimizing contributions from paraffin-filled holes. The correlation curves are shown in *Appendix 2—figure 1*. The intersection of the curve with the half-bit threshold yields the resolution estimate, indicated with dashed black line. Correspondingly, a half-period resolution of $0.71\,\mu\mathrm{m}$ and $0.39\,\mu\mathrm{m}$ (or better) is obtained for the parallel and cone beam dataset, respectively. However, since the splitted dataset resulted in only 721 angles for the reconstruction, i.e. the resolution estimate is severely affected by under-sampling artifacts, and hence can only serve as an upper bound. *Appendix 2—figure 1* illustrates the analysis.

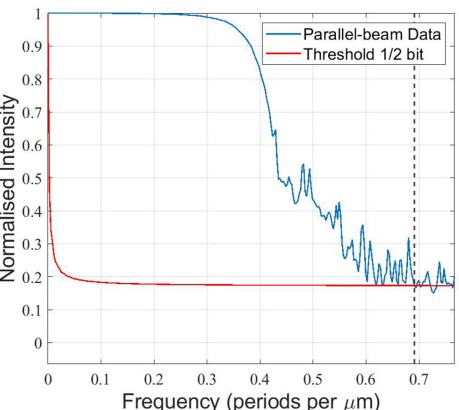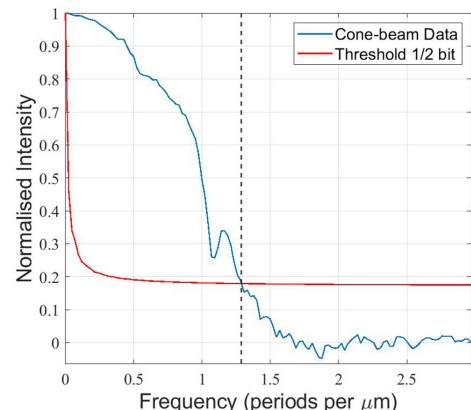

**Appendix 2—figure 1.** Quantification of the 3D-resolution for the example of sample V. FSC analysis was carried out for two independent reconstructions, each from half the projections of the scan.

## Appendix 3

### Further datasets

Controls: The imaging workflow was also applied to hydrated and/or healthy lung tissue as a control. First, overview scans covering the entire samples were recorded. Then, hydrated 1 mm biopsy punches (two for CTRLI, one for CTRLII and CTRLIII, where CTRLII and CTRLIII are from the same patient) were recorded in (1 - parallel beam) configuration. Biopsy punches from CTRLII and CTRLIII were also examined in (2 - cone beam) mode. *Appendix 3—figure 1* presents (a) the rendering of the hydrated control, and (b,c) virtual slices through the reconstruction volume, showing the lung parenchymal architecture in healthy control tissue. UA-stained samples: For all Covid-19 patients, autopsies were also treated by UA-staining in order to increase the contrast. Stitched overview scans in (1 - parallel beam) configuration were recorded, similar to *Figure 3*. From the UA-labeled tissue blocks of patients I, III, IV and V, 1 mm, biopsy punches were than scanned in the same configuration (*Figure 4*). Using the (2 - cone beam) setup, these samples from I, III and IV were imaged at $8.0\,\mathrm{keV}$ x-rays, while V was examined at $13.8\,\mathrm{keV}$, as shown for *Figure 5*. Variation of propagation distance: Scans of the unstained tissue block from patient II were recorded at different propagation distances ($z_{12} = 50, 100$ and $125\,\mathrm{keV}$) and different x-ray energies ($13.3, 13.8, 14.3$ and $14.8\,\mathrm{keV}$). In cone-beam configuration, the unstained biopsy punch from patient I was scanned at $13.8\,\mathrm{keV}$ x-rays. Compact $\mu$CT scans: Prior to the synchrotron experiment, some of the samples have been examined with a laboratory phase-contrast $\mu$CT-setup in large mm$^2$-sized FOV-configuration (Liquid metal jet source, $\mathrm{K}_\alpha = 9.5\,\mathrm{keV}$, $px_{\mathrm{eff}} = 5\,\mu\mathrm{m}$, $z_{12} = 1.7\,\mathrm{m}$, 1200 projections of 1 s exposure time with a flat panel CMOS detector with $150\,\mu\mathrm{m}$ Gadox-scintillator, PerkinElmer, USA) (*Bartels et al., 2013*). Metal-staining (here UA) of the lung tissue helped to achieve sufficient contrast, similar to previous $\mu$CT-studies of other biological tissues (*Müller et al., 2017*; *Busse et al., 2018*; *De Clercq et al., 2019*). The resulting overview scans could also be correlated well with histological sections.

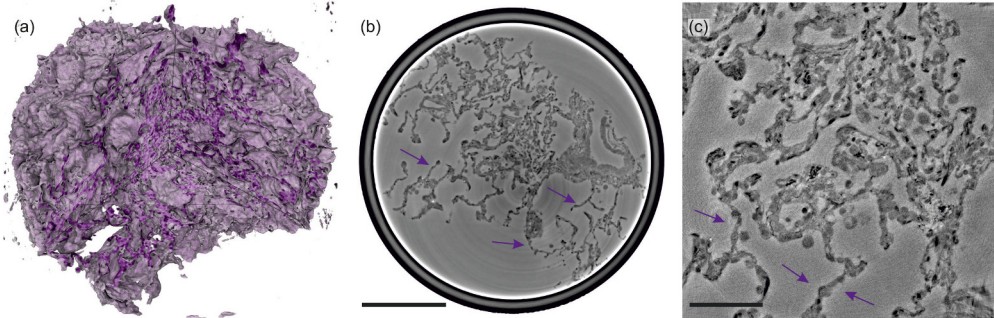

**Appendix 3—figure 1.** Illustrations of control lung tissue (hydrated). (**a**) Volume rendering of the tissue block ($0.97 \times 1.00 \times 0.73$ mm$^3$) and (**b**) slice through the volume, examined in PB-configuration. (**c**) Slice from the cone-beam scan, arrows indicating the structure of a healthy septum. Particularly, macrophages and erythrocytes emerge. Scale bars: (**b**) $100\,\mu\mathrm{m}$ and (**c**) $300\,\mu\mathrm{m}$.

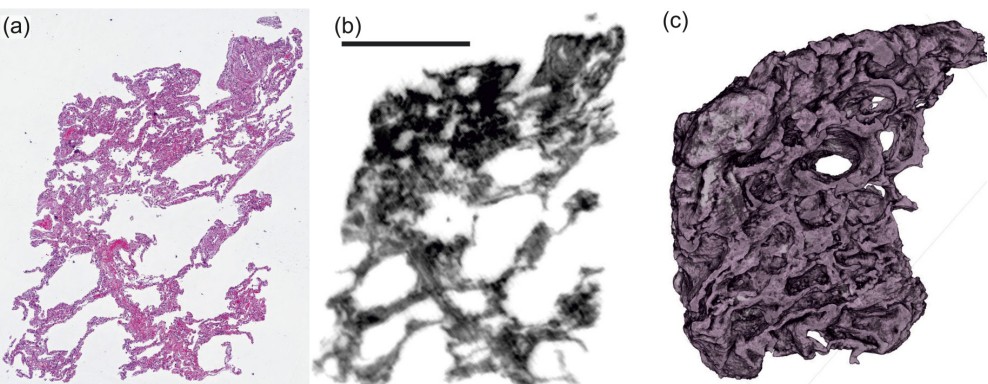

**Appendix 3—figure 2.** Screening with a laboratory phase-contrast μCT-setup. (UA-stained tissue

block, patient I). (**a**) Histological and (**b**) correlative virtual slice from laboratory phase-contrast tomography. (**c**) Volume rendering of the entire tissue block from a similar perspective. Scale bar: 1 mm.

