## [Decision Letter]

**Decision letter after peer review:**

Thank you for submitting your article "3d Virtual Pathohistology of Lung Tissue from Covid-19 Patients based on Phase Contrast X-ray Tomography" for consideration by *eLife*. Your article has been reviewed by a Senior Editor, a Reviewing Editor, and three reviewers. The following individuals involved in review of your submission have agreed to reveal their identity: Ashraf Khan (Reviewer #1); Kohkan Shamsi (Reviewer #2).

The reviewers have discussed the reviews with one another and the Reviewing Editor has drafted this decision to help you prepare a revised submission.

Summary:

This is novel technology where the authors describe the tomographic imaging of lung biopsies from six Covid-19 patients. For this purpose, hard X-ray phase-contrast imaging at P10 beamline together with X-ray optics has been utilized, with the local tomography data reconstructed and analyzed using state-of-the-art algorithms and tools.

Essential revisions:

1) The authors are encouraged to comment on the resolution of images that is not ideal (most likely they are autopsy samples). For example, in Figure 2 in Appendix 1—figure 2, area marking vascular thrombus is not unequivocally diagnostic of a thrombus at the submitted magnification and fibrin cannot be definitively recognized within the vascular lumen. If possible, the authors should reproduce findings in a limited way on surgically resected samples, which will likely yield better histological resolution for comparative analysis.

2) The paper would benefit from a direct comparison of hard X-ray phase tomography and histology. Because tomography is essentially non-destructive, the authors could prepare the related histological slides. The basis for this could be Appendix 1—figure 2. The authors should identify which information could be extracted only from one technique to better demonstrate the complementarity of the two techniques.

3) The Introduction, noted as a patchwork, requires considerable improvement. It does not contain a clear hypothesis nor a scientific question. It is difficult to understand the true focus of the work.

---

## [Author Response]

Essential revisions:1) The authors are encouraged to comment on the resolution of images that is not ideal (most likely they are autopsy samples). For example, in Figure 2 in Appendix 1—figure 2, area marking vascular thrombus is not unequivocally diagnostic of a thrombus at the submitted magnification and fibrin cannot be definitively recognized within the vascular lumen. If possible, the authors should reproduce findings in a limited way on surgically resected samples, which will likely yield better histological resolution for comparative analysis.

We have followed this suggestion and have critically commented on the resolution. In order to provide quantitative resolution estimates we have performed Fourier Shell Correlation analysis (after splitting the datasets into two) and have included this analysis in new Appendix 2 with a figure showing the FSC curves and corresponding text. The newly added exact comparison between a 2d histology and the corresponding virtual section through the reconstruction volume, as provided in response to point 2 below and shown in Appendix 1—Figure 3, will also help the reader to form an opinion concerning resolution by visual comparison. We agree that autopsy tissue may also suffer from structural alterations. Unfortunately, in contrast to common lung tumor interventions, we do not have comparable access to surgically resected samples of Covid-19 patients at this point.

2) The paper would benefit from a direct comparison of hard X-ray phase tomography and histology. Because tomography is essentially non-destructive, the authors could prepare the related histological slides. The basis for this could be Appendix 1—figure 2. The authors should identify which information could be extracted only from one technique to better demonstrate the complementarity of the two techniques.

In response to this suggestion by the reviewer(s), we have performed histological analysis of samples which were previously scanned, immediately after obtaining the report. Since this requires sectioning of the 1mm biopsy punch, the sample had to be re-embedded in a larger paraffin block, sectioned, stained and imaged. We are very glad that we can now provide exactly the type of comparison suggested by the reviewer, presented as a new Figure in the Appendix (Appendix1—figure 3), as suggested. We have identified the same plane in the 3d CT data, and now can show blood vessels with thrombi side by side in the two modalities. The particular advantage of 3d histology is that the thrombi can be followed from plane to plane such that the number and size of thrombi in a single vessel can be quantified. 2d histology, on the other hand, informs us more precisely on the biomolecular origin of the structure via the color shades of the stained structure which can be interpreted based on correlation with previous studies. We assert, however, that quantitative analysis of tissue structure requires the full dimensionality, and ideally isotropic resolution. Loosely speaking, 3d histology compares to 2d histology, as would (3d) anatomy to a putative 2d anatomy. The relevance of dimensionality is by now well appreciated now in different life science research fields: three-dimensional (3d) cell culture systems have gained increasing interest, light sheet microscopy in combination with clearing offers 3d views of small organisms. At the same time, we know that in the end, many questions will still be answered by 2d histology, in the same way as diagnosis in radiology is still largely based on 2d radiography , and will involve CT scans only when needed. The most important advantage of 2d histology is the fact that information on specific proteins can be obtained, while immuno-labeling with x-ray contrast agents is not yet within reach!

We have added a comparison stressing the complementarity of the approaches in the paragraph describing the new figure in the Appendix, and also address this in the revised Discussion section.

3) The Introduction, noted as a patchwork, requires considerable improvement. It does not contain a clear hypothesis nor a scientific question. It is difficult to understand the true focus of the work.

We have completely rewritten paragraph three and four of the Introduction in response to this request. We now write in the new key sentences opening paragraph three:

“In this work, we want to demonstrate the potential of propagation-based phase contrast x-ray tomography as a tool for virtual 3d histology in general, and in particular for the histopathology of Covid-19. Our work is based on the assertion that integration of 3d morphological information with well established histology techniques can provide a substantial asset for unraveling the pathophysiology of SARS-Cov-2 infections. To this end, we have collected x-ray tomography data from the same autopsies, which were previously studied by immunohistochemical analysis and measurements of gene expression (Ackermann et al., 2020). We ask in particular, whether DAD and the morphology of blood vessels can be visualized and quantified in 3d. This is a timely scientific question in Covid-19 research, especially view of the findings of increased intussusceptive angiogenesis reported in (Ackermann et al., 2020). Here we present first results obtained from the postmortem lung samples of…”